

# Self-reported body weight and weight-related stigmatization experiences among young adult women—two contexts, but similar attitudes related to body image, mental *self*-schemas, self-esteem, and stereotypes of people with obesity

Łukasz Jach[1] and Sonia Krystoń[2]

[1] Institute of Psychology, Faculty of Social Sciences, University of Silesia, Katowice, Poland
[2] Faculty of Social Sciences, University of Silesia, Katowice, Poland

Corresponding author
Łukasz Jach, lukasz.jach@us.edu.pl

## ABSTRACT

**Background:** Weight stigma is a serious challenge because of its negative impact on human health and harmful psychological and behavioral consequences. The aim of the study was to explore and compare the relationships between self-reported body weight and weight-related stigmatization experiences and body image, mental self-image, self-esteem, and stereotypes concerning people with obesity among young adult Polish women ($N$ = 374; aged between 18 and 35).

**Methods:** The study was conducted online on a Polish sample recruited through a social network site, a website, and snowball sampling. Body mass index (BMI) was used to assign the respondents to groups with normal or excess weight. We tested whether women enrolled in the study experienced weight-related stigmatization using two questions based on the concepts of spoiled identity and related to the obesity stigma. The Contour Drawing Rating Scale was used to study different aspects of the body image and discrepancies between them. The Self-Discrepancy Questionnaire was used to study the self-schemas associated with mental qualities. The Polish version of the Rosenberg's Self Esteem Scale was applied to determine self-esteem level. Stereotypes concerning people with obesity were studied using the semantic differential method.

**Results:** Although excess weight was associated with weight-related stigmatization experiences, many women reported confronting such stigmatization even though their body weight was normal according to the World Health Organization (WHO) standards. Women with excess weight and women with weight-related stigmatization experiences were characterized by larger discrepancies between the *actual* body image and the *ideal*, *reflected*, and *ought* body image, lower self-esteem, and more negative beliefs about their mental *actual* and *reflected self* compared to women with normal weight and without weight-related stigmatization experiences. The study participants from all groups tended to believe their *actual* body image to be ampler than the *ideal* and the *ought* body images. They also believed that other people perceived their mental qualities more positively than they did. The study groups were also characterized by negative stereotypes of people with obesity,

although these stereotypes were more vital in women with excess weight and women who experienced weight-related stigmatization.

**Conclusion:** The study shows the similarity between psychological functioning of women with self-reported excess weight and those who experience weight-related stigma. The results also provide guidelines for practical actions aimed at reducing negative mental outcomes associated with not conforming to body weight standards.

## INTRODUCTION

Weight-related stigma can be defined as the experience of verbal or physical abuse secondary to being overweight or obese (*Wu & Berry, 2017*). Stigmatization signals may come, for instance, from employers and co-workers (*Watson, Levit & Lavack, 2017*), scholars (*Cameron, 2016*), health care employees ((*Tomiyama et al., 2018*), or family members and partners (*Puhl & Brownell, 2001*; *Hunger et al., 2015*). Weight-related stigma has a negative impact on health (*O'Brien et al., 2016*; *Wu & Berry, 2017*), psychological well-being (*e.g.*, reduced self-esteem, anxiety, depression, eating disorders; see *e.g.*, *Kahan & Puhl, 2017*; *Walsh et al., 2018*) or everyday functioning (*e.g.*, maladaptive coping strategies, binge-eating or exercise avoidance; see: *Hayward, Vartanian & Pinkus, 2018*; *Jackson & Steptoe, 2017*; *Wu & Berry, 2017*). Stigmatization causes psychological and physiological stress, depletes self-control resources necessary to regulate healthy behavior, and increases motivation to avoid potentially discriminatory situations (*Ashmore et al., 2008*; *Vani et al., 2020*; *Hayward, Vartanian & Pinkus, 2018*; *Hunger et al., 2015*; *Jackson & Steptoe, 2017*). Experiencing constant stigma weakens mental and physical health and contributes to weight gain, perpetuating the source of the stigma (*Puhl & Brownell, 2001*, *2006*). By experiencing long-term stigmatization in various social situations, individuals with excess weight may also internalize the stigma (*Kahan & Puhl, 2017*; *Pearl & Puhl, 2016*) and develop a negative body image.

The pressure exerted on individuals with excess weight is a socially acceptable form of stigmatization (*Kasardo & McHugh, 2015*; *Myers & Rothblum, 2010*; *Puhl & Brownell, 2001*; *Puhl & Heuer, 2010*), and the negative attitude is all the stronger because the stigmatized trait seems to be controllable (*Crandall & Biernat, 1990*; *Kasardo & McHugh, 2015*; *Puhl & Heuer, 2010*; *Vartanian, 2010*). People with obesity are perceived as lazy, ugly, lonely, clumsy, and weak-willed (*Davison & Birch, 2004*; *Głębocka & Szarzyńska, 2005*; *van Leeuven, Hunt & Park, 2015*). There is a stereotypical erroneous belief that experiencing shame due to excess body weight stimulates the individual to change their eating habits, thus bringing the desired effects (*Hunger et al., 2015*; *Powell & Hendricks, 1999*).

A concept related to the sense of lacking the desired qualities is the regulatory factor in the self-discrepancy theory by *Higgins (1987*, *1989)*. According to this approach, the structure of the human *self* consists of three elements. The *actual self* represents how

individuals perceive themselves; the *ideal self* refers to what kind of person they would like to be, while the *ought self* refers to the qualities they think they should have to meet external expectations. Referring to the theory by *Goffman (1986)*, the *reflected self* can also be indicated. It is a personal conviction about the traits other people attribute to the individual. Discrepancies between the schemas affect how individuals treat themselves. A negative difference between the *actual self* and the *ought self* generates a feeling of failure to meet the standards associated with shame, self-contempt, and the desire to punish themselves. Inconsistency between the *actual self* and the *reflected self* induces problems with defining one's own identity and difficulties related to defining one's aspirations. However, a negative mismatch between the *actual self* and the *ideal self* is connected with a depressed mood and a feeling of self-disappointment (*Bąk, 2002*).

Research shows that body weight is also associated with self-esteem, although the study results are ambiguous. Longitudinal studies by *Kiviruusu et al. (2016)* showed that in women body weight and self-esteem were negatively related regardless of their age, but negative relations between these variables characterized middle-aged men only. Similarly, a meta-analysis by *Miller & Downey (1999)* showed that while the overall relationship between body weight and self-esteem was negative, this effect was more substantial for women. Self-esteem is the individual's belief related to one's general value, which may be expressed in verbal declarations (*Rosenberg, 1965*; *Łaguna, Lachowicz-Tabaczek & Dzwonkowska, 2007*). Although it is commonly believed that people with low self-esteem are convinced of their worthlessness, their self-value is rather unstable. Their self-esteem is thus more malleable and susceptible to new information (*Wojciszke, 2011*). This approach is in line with *Goffman (1986)*, who noted that stigmatized individuals displayed an ambivalent attitude toward themselves and people bearing the same stigma. The related stereotypes provide the context which makes it possible to perceive excess weight in terms of the overall stigma (*e.g.*, *Watson, Levit & Lavack, 2017*). Stereotypes have an underlying sociocultural foundation and create an environment that shapes how people treat one another (*Link & Phelan, 2001*). The stereotyped individuals may tend to share the stereotype or behave in accordance with it (*Latrofa, Vaes & Cadinu, 2012*). Moreover, such individuals tend to stigmatize other bearers of the same stigma, depending on how visible it is (*Becker & Tausch, 2014*).

## Aims of the study and research hypotheses

The aim of the study was to explore and compare the relationships between self-reported body weight and weight-related stigmatization experiences and body image, mental self-image, self-esteem, and stereotypes concerning people with obesity among young adult women (aged 18–35). The above variables may be shaped in close connection with body weight and provide the context for functioning in such areas of everyday life as social relations, work, education, manners of relaxation, and spending free time.

The study focused on young adult women due to several reasons. Women perceive their body weight as higher than it actually is more often than men (*Chang & Christakis, 2003*). They are also more dissatisfied with their weight and body image (*Rozin & Fallon, 1988*). In women, excess body weight is also related to more pronounced negative

psychological consequences (*Magallares & Pais-Ribeiro, 2014*; *Miller & Downey, 1999*). Focusing on the group of young adult women was related to developmental, psychological, and sociocultural factors, the specificity of which in each developmental period may be differently related to both the perception of body weight and the studied variables (*Hutchison, Leigh & Wagner, 2016*).

Since weight stigma is based not only on the actual appearance of the body but also on subjective feelings, both the self-reported BMI and weight-related stigmatization experiences were considered. It was hypothesized (H1) that self-reported normal or excess weight of the studied women would not coincide with their weight-related stigmatization experiences.

Since excess weight is considered an undesirable trait, another hypothesis (H2) was formulated, *i.e.*, women with excess weight would have lower self-esteem than women with normal weight, and women experiencing weight-related stigmatization would have lower self-esteem than those not experiencing such stigmatization.

Considering the significance of sociocultural influences for the attitude toward one's body, we hypothesized (H3) that compared to women with normal weight and women not experiencing weight-related stigmatization, women with excess weight and women experiencing weight-related stigmatization (a) in terms of the *actual* body image and the mental *actual self* would tend to perceive themselves more distinctly as "fat persons" and have a less positive image of their mental traits. Additionally, (b) in terms of the *ideal* and *ought* body image, they would display similar schemas related to the preferred figure. Next, (c) in terms of the *reflected* body image and the mental *reflected self*, they would tend to recognize that other people perceive them as "fat persons" and as individuals with less positive mental traits. Finally, (d) they would display larger discrepancies between *actual* body image and other aspects of the body image as well as more discrepancies between the mental *actual* and *reflected self*.

Concerning the potential internalization of weight-related stereotypes and the occurrence of lack of identification with the stigmatized group, we also hypothesized that (H4) women with excess weight and women experiencing weight-related stigmatization would show a similar tendency to maintain the stereotypes of people with obesity compared to women with normal body weight and women not experiencing weight stigma.

## MATERIALS & METHODS

### Participants and procedure

The study was conducted online on a Polish sample of female participants.
The participants were recruited through social network site (Facebook: profile of one of the authors of the manuscript and profile *Kobieta Puszysta*, dedicated to women with excess weight), a website (https://www.kobietaxl.pl; a Polish website dedicated to women with excess weight) and snowball sampling (people who received information about the study were asked to pass it on to others *via* the same social networking site). The criteria for inclusion in the sample were: being a woman between 18 and 35 years of age, having normal or excess BMI, using the Polish language, and having access to the Internet.

The procedure was anonymous and voluntary. It was approved by the Ethics Committee of the University of Silesia (decision number: KEUS.18/04.2020). The participants were informed that the research would be related to the psychological aspects of experiencing stigma and gave their informed consent to participate by going to the subsequent sections of the survey. To give the respondents a greater sense of anonymity, signed consent forms to participate in the study were not collected from them. However, each respondent had the opportunity to withdraw from the research procedure at any time. Additionally, the participants had the opportunity to contact the study authors in case of any questions or comments.

Four hundred seventy-five women responded. However, the data from 374 individuals who met the inclusion criteria were used for further analyses. Fifty-two individuals with BMI values indicating underweight, two individuals under 18 years of age, 45 individuals over 35 years of age, and two respondents who did not give information about their body weight were excluded from the study. No information on ethnicity was collected during the study. However, since the study was conducted in the Polish language, it can be assumed that the vast majority of respondents were Caucasian women from Poland or other Slavic countries where Polish is relatively often used.

Descriptive statistics related to demographic aspects and psychological variables are presented in Table 1. Due to the non-normality of the distributions of the measured continuous variables, we only present the values of the medians and the lower and upper quartiles.

## Data collection measurements

Body Mass Index (BMI) was used to assign the respondents to groups with self-reported normal or self-reported excess weight. According to the World Health Organization (WHO), normal BMI ranges from 18.50 to 24.99, BMI between 25.00 and 29.99 indicates overweight, and the values of 30.00 and more indicate obesity (URL: https://www.who.int/data/gho/data/themes/topics/topic-details/GHO/ncd-risk-factors; accessed on 7 April 2021). In line with the WHO standards, BMI ranging between 18.50 and 24.99 indicated normal body weight, while higher BMI values indicated excess weight.

We used two questions, based on the concept of spoiled identity (*Goffman, 1986*) and concepts related to the obesity stigma (*Alegría & Larsen, 2015*; *Davison & Birch, 2004*; *Hunger et al., 2015*) to test whether the participants experienced weight-related stigmatization. The following questions were asked: "Do you think you are fat now?"; and "Are you currently ashamed of how much you weigh?". If the participant answered affirmatively to either of these questions, she was assigned to the group with weight-related stigmatization experiences (For a discussion of the validity of the adopted method of measuring weight-related stigma experiences, see the *Study limitations* section.). Therefore, each participant was assigned to two categories related to (normal or excess) BMI and (the presence or absence of) weight-related stigmatization experiences.

The Contour Drawing Rating Scale (*Thompson, 2004*; *Wertheim, Paxton & Tilgner, 2004*) was used to study different aspects of the body image and their discrepancies. The tool contains nine female figure drawings arranged on a scale from very slim to very
Table 1 Descriptive statistics of quantitative and qualitative variables in the study.

| Quantitative variable | Median | Lower quartile | Upper quartile | Shapiro–Wilk W | Shapiro–Wilk p-value |
|---|---|---|---|---|---|
| Age (y) | 22.00 | 21.00 | 24.00 | 0.895 | <0.001 |
| Height (m) | 1.67 | 1.62 | 1.70 | 0.983 | <0.001 |
| Self-reported body weight (kg) | 60.00 | 55.00 | 69.00 | 0.814 | <0.001 |
| BMI | 21.72 | 20.20 | 24.24 | 0.748 | <0.001 |
| (1) *Actual* body image | 5.00 | 4.00 | 7.00 | 0.956 | <0.001 |
| (2) *Ideal* body image | 4.00 | 3.00 | 4.00 | 0.924 | <0.001 |
| (3) *Reflected* body image | 5.00 | 4.00 | 6.00 | 0.957 | <0.001 |
| (4) *Ought* body image according to men | 4.00 | 3.00 | 5.00 | 0.910 | <0.001 |
| (5) *Ought* body image according to women | 3.00 | 2.00 | 3.00 | 0.912 | <0.001 |
| (1)–(2) Discrepancy | 2.00 | 1.00 | 3.00 | 0.940 | <0.001 |
| (1)–(3) Discrepancy | 0.00 | 0.00 | 1.00 | 0.873 | <0.001 |
| (1)–(4) Discrepancy | 2.00 | 1.00 | 3.00 | 0.963 | <0.001 |
| (1)–(5) Discrepancy | 2.50 | 2.00 | 4.00 | 0.960 | <0.001 |
| (6) Mental *actual* self | 81.00 | 69.00 | 92.00 | 0.972 | <0.001 |
| (7) Mental *reflected* self | 87.00 | 74.00 | 98.00 | 0.975 | <0.001 |
| (6)–(7) Discrepancy | −6.00 | −12.00 | 0.00 | 0.972 | <0.001 |
| Self-evaluation | 28.00 | 25.00 | 32.00 | 0.991 | 0.019 |
| Stereotype of people with obesity | 2.30 | 1.80 | 2.80 | 0.956 | <0.001 |
| **Qualitative variable** | **Yes** | **% Yes** | **No** | **% No** | |
| Weight norm (BMI according to the WHO criteria) | 298 | 78.6 | 80 | 21.4 | The group of participants with excess weight includes 53 participants with overweight and 27 participants with obesity. |
| "Do you think you are currently fat?" | 129 | 34.5 | 245 | 65.5 | |
| "Are you currently ashamed of how much you weigh? | 154 | 41.2 | 220 | 58.8 | |
| Overall weigh-related stigma | 173 | 46.3 | 201 | 53.7 | |

obese. Each respondent was asked to indicate the figure showing (1) what she would like to look like (*ideal* body image), (2) what she looked like in her opinion (*actual* body image), (3) what she looked like in the opinion of others (*reflected* body image), (4) the type most preferred by men (*ought* body image according to men) and (5) the type most preferred by women (*ought* body image according to women).

To study the *self*-schemas associated with mental qualities, we used the Self-Discrepancy Questionnaire (SkRAP; see: *Wojdyło & Buczny, 2011*). This tool consists of twelve items referring to such positively valued traits as initiative, cleverness, creativity, wisdom, motivation, courage, diligence, prudence, strong will, orderliness, perseverance, and resourcefulness. The task of the respondent was to indicate the extent to which she possessed the relevant quality (mental *actual self*) and the extent to which the trait was seen in her by other people (mental *reflected self*). The responses were assessed on a scale from 1 (very low level) to 10 (very high level). The comparison between the overall results for the mental *actual self* and the mental *reflected self* allowed the evaluation of

discrepancies between them. The Cronbach's α coefficients were 0.87 for the mental *actual self* scale and 0.89 for the mental *reflected self* scale.

The Polish version of the Rosenberg's Self Esteem Scale (*Łaguna, Lachowicz-Tabaczek & Dzwonkowska, 2007*) was used to determine self-esteem levels. It consists of ten items related to the respondents' beliefs about themselves. They refer to each item by selecting one of four answers arranged on a scale from "strongly agree" to "strongly disagree". The higher the overall score, the higher the respondent's self-esteem is. The Cronbach's α coefficient was 0.89.

Stereotypes about individuals with obesity were studied using the semantic differential method (*Głębocka & Szarzyńska, 2005*; *Rosenberg & Navaro, 2018*). The task of the respondents was to indicate what they believed other people thought about individuals with obesity in relation to ten opposite pairs of terms such as "lazy/hard-working", "scruffy/neat", "messy/tidy", "lacking control/strong-willed", "ugly/pretty"; "stupid/smart", "self-indulgent/disciplined", "not setting limits for themselves/self-restricting", "unattractive/attractive" and "quitting/persistent". The respondents indicated their answers on a seven-point scale. Values 1 to 3 indicate a tendency toward negative categories, while values 5 to 7 indicate a positive categorization. The overall result is the average score obtained for all the items, ranging from 1 to 7. The Cronbach's α coefficient was 0.88.

## Data analysis methodology

Statistica 13 and JASP 0.14.1.0 were used for statistical analysis. We decided to use the McNemar chi-square test to test hypothesis 1. This test is used with matched pairs of dichotomous measurements to check that the frequencies of certain conditions are equal. In the case of hypothesis 1, excess weight was written in rows, and having weight-related stigmatization experiences was written in columns. We planned to use two-sample comparison tests to test hypotheses 2–4. Additionally, during the analyses related to hypotheses 2 and 3, we decided to make a series of comparisons of the results obtained by the same people on different scales; therefore we planned to use tests for matched pairs.

In order to check the possibility of using parametric tests for the planned analyses, we initially conducted an analysis of the distributions of the residuals. Due to the numerous noticed deviations of the residuals from the normal distribution, we decided to use non-parametric tests.

Due to the non-normality of distribution of the studied quantitative variables (see Table 1), we used non-parametric tests, *i.e.*, the McNemar chi-square test, the Mann–Whitney $U$ test (for two-sample comparisons), and the Wilcoxon signed-rank test (for matched pairs). A two-sided $p$-value below 0.05 was considered statistically significant. For multiple comparisons carried out in connection with hypotheses 2 and 3, the Holm–Bonferroni correction was applied. The corrected $p$-values were calculated for all combined analyzes performed in accordance with hypotheses 2 and 3. Similarly, the corrected $p$-values were calculated for the combined set of additional analyzes. Rank biserial correlation coefficients and matched rank biserial correlation coefficients were used to determine the effect size.
**Table 2 Body image, self-esteem and mental *self*-schemas in the groups with different BMI and weight-related stigmatization experiences—Mann–Whitney *U* tests.**

| Variables | Median in the BMI group (interquartile range) | | | | Median in the weight-related stigmatization experiences group (interquartile range) | | | |
|---|---|---|---|---|---|---|---|---|
| | Normal | Excess | *p* | ES | No | Yes | *p* | ES |
| (1) *Actual* body image | 5.0 (2.0) | 8.0 (1.0) | **<0.001** | −0.822 | 5.0 (1.0) | 6.0 (2.0) | **<0.001** | −0.599 |
| (2) *Ideal* body image | 4.0 (1.0) | 4.5 (1.5) | **<0.001** | −0.399 | 4.0 (1.0) | 4.0 (2.0) | 0.264 | −0.064 |
| (3) *Reflected* body image | 4.0 (1.0) | 7.0 (1.5) | **<0.001** | −0.819 | 4.0 (2.0) | 6.0 (2.0) | **<0.001** | −0.604 |
| (4) *Ought* body image according to men | 4.0 (2.0) | 4.0 (2.0) | 0.787 | −0.019 | 4.0 (2.0) | 3.0 (1.0) | 0.072 | 0.104 |
| (5) *Ought* body image according to women | 3.0 (1.0) | 3.0 (1.5) | 0.871 | 0.011 | 3.0 (1.0) | 3.0 (1.0) | 0.087 | 0.098 |
| Discrepancy (1)–(2) | 1.0 (1.0) | 3.0 (2.0) | **<0.001** | −0.635 | 1.0 (2.0) | 2 (1.0) | **<0.001** | −0.635 |
| Discrepancy (1)–(3) | 0.0 (1.0) | 0.0 (1.0) | **0.003** | 0.202 | 0.0 (1.0) | 0 (1.0) | **0.008** | 0.147 |
| Discrepancy (1)–(4) | 1.0 (1.0) | 4.0 (2.0) | **<0.001** | −0.739 | 1.0 (2.0) | 3 (2.0) | **<0.001** | −0.612 |
| Discrepancy (1)–(5) | 2.0 (2.0) | 5.0 (2.0) | **<0.001** | −0.730 | 2.0 (2.0) | 4 (3.0) | **<0.001** | −0.595 |
| (6) Mental *actual* self | 82.0 (21.0) | 76.5 (22.5) | **0.002** | 0.228 | 84.0 (18.0) | 77.0 (23.0) | **<0.001** | 0.242 |
| (7) Mental *reflected* self | 88.0 (22.0) | 83.0 (22.0) | 0.011 | 0.184 | 91.0 (22.0) | 83.0 (24.0) | **0.001** | 0.190 |
| Discrepancy (6)–(7) | −6.0 (12.0) | −6.0 (17.0) | 0.874 | −0.012 | −6.0 (12.0) | −6.0 (15.0) | 0.642 | 0.028 |
| Self-esteem | 29.0 (7.0) | 27.0 (7.0) | 0.012 | 0.183 | 30.0 (6.0) | 26.0 (8.0) | **<0.001** | 0.396 |

Note:
  *p*-value significant with the Holm–Bonferroni correction were presented in bold; ES, effect size is given by the rank biserial correlation.

## RESULTS

### H1: Self-reported body weight and experience of weight-related stigmatization

The McNemar chi-square test was used to study the relationship between the actual body weight and weight-related stigmatization experiences among participants. Its result showed a significant relationship between the studied variables at $p < 0.001$. Out of 80 participants with excess weight, only eight (10.0%) did not declare weight-related stigmatization experiences. However, of 173 participants with weight-related stigmatization experiences, the majority (101 participants; 58.4%) had normal BMI according to the WHO standard. The obtained results supported hypothesis 1, positing inconsistency between the self-reported weight and weight-related stigmatization experiences. The experiences of weight-related stigmatization occurred much more frequently among the respondents than excess weight as indicated by BMI values.

### H2 and H3: Body image, self-esteem, and mental self-schemas in the context of self-reported weight and weight-related stigmatization experiences

The groups with different BMI values and weight-related stigmatization experiences were compared in terms of body image, self-esteem, and self-schemas related to mental traits. The relevant information is given in Table 2. Due to multiple comparisons, the Holm–Bonferroni correction was used (the number of planned comparisons was 13 for

groups with different BMI and 13 for groups with different weight-related stigmatization experiences).

Significant differences between the groups were similar, regardless of whether they were distinguished due to self-reported body weight or weight-related stigmatization experiences. As far as the *actual* body image was concerned, the respondents with excess BMI indicated ampler figures compared to respondents with normal weight. Also, the respondents with weight-related stigmatization experiences indicated ampler figures compared to respondents without such experiences. Significant differences were also observed between women who differed in terms of BMI regarding the *ideal* body image, *i.e.*, women with normal weight indicated slimmer figures as desirable. Regardless of the criterion, significant differences were found between the groups regarding the *reflected* body image, which indicated ampler figures by respondents with excess weight and respondents with weight-related stigmatization experiences. No differences were found between the groups in terms of the aspects of the *ought* body image, which indicated the similarity of figures that were considered preferred by other women and men.

The discrepancies between the *actual* body image and other aspects of body image were also analyzed. They became more prominent in women with excess weight than in those with normal weight and in women with weight-related stigmatization experiences than in those without such experiences.

The groups also differed in terms of mental aspects. The respondents without weight-related stigma experiences had higher self-esteem and a more positive image of mental traits compared to those with weight-related stigma experiences. However, the group with normal BMI values showed only a higher level of the mental *actual self* than the group with excess BMI. No significant differences were found in terms of the discrepancy between the mental *actual self* and the mental *reflected self* between the groups.

The results partly confirmed hypothesis 2. Women with weight-related stigmatization experiences reported lower self-esteem than women without such experiences. However, the results showed no significant differences in self-esteem in women with normal and excess BMI.

Hypothesis 3 was also only partly confirmed. Regardless of the criterion, women with excess weight and those with weight-related stigma experiences indicated ampler figures corresponding to their *actual* body image, and they had a less positive image of their mental traits. In most aspects, they were also characterized by larger discrepancies between the *actual* body image and other aspects of the body image. A result that diverged from the hypothesis was the difference in terms of the body figure considered ideal. Women with excess BMI values preferred slightly ampler figures than women with normal weight. Another result that was inconsistent with the hypothesis was the comparable level of numerical discrepancies in the participants between the mental aspects of the *actual self* and the *reflected self*. However, positive values of rank-biserial correlation coefficients indicated a larger discrepancy between the *actual* body image and the *reflected* body image, characterizing women with normal BMI and women without weight-related stigmatization experiences rather than women with excess BMI and with weight-related stigmatization experiences.

**Table 3 Differences between body image aspects and mental *self* aspects in the groups with different BMI and different weight-related stigmatization experiences—Wilcoxon signed-rank tests.**

| Compared variables | Median in the BMI group (interquartile range) | | Median in the weight-related stigmatization experiences group (interquartile range) | |
|---|---|---|---|---|
| | Normal | Excess | No | Yes |
| *Actual* body image | 5.0 (2.0) | 8.0 (1.0) | 5.0 (1.0) | 6.0 (2.0) |
| *Ideal* body image | 4.0 (1.0) | 4.5 (1.5) | 4.0 (1.0) | 4.0 (2.0) |
| | $p < 0.001$ | $p < 0.001$ | $p < 0.001$ | $p < 0.001$ |
| | ES = 0.926 | ES = 1.000 | ES = 0.905 | ES = 0.978 |
| *Actual* body image | 5.0 (2.0) | 8.0 (1.0) | 5.0 (1.0) | 6.0 (2.0) |
| Reflected body image | 4.0 (1.0) | 7.0 (1.5) | 4.0 (2.0) | 6.0 (2.0) |
| | $p < 0.001$ | $p = 0.183$ | $p < 0.001$ | $p < 0.001$ |
| | ES = 0.705 | ES = 0.220 | ES = 0.773 | ES = 0.410 |
| *Actual* body image | 5.0 (2.0) | 8.0 (1.0) | 5.0 (1.0) | 6.0 (2.0) |
| *Ought* body image according to men | 4.0 (2.0) | 4.0 (2.0) | 4.0 (2.0) | 3.0 (1.0) |
| | $p < 0.001$ | $p < 0.001$ | $p < 0.001$ | $p < 0.001$ |
| | ES = 0.827 | ES = 1.000 | ES = 0.724 | ES = 0.986 |
| *Actual* body image | 5.0 (2.0) | 8.0 (1.0) | 5.0 (1.0) | 6.0 (2.0) |
| *Ought* body image according to women | 3.0 (1.0) | 3.0 (1.5) | 3.0 (1.0) | 3.0 (1.0) |
| | $p < 0.001$ | $p < 0.001$ | $p < 0.001$ | $p < 0.001$ |
| | ES = 0.971 | ES = 1.000 | ES = 0.960 | ES = 0.996 |
| Mental *actual self* | 82.0 (21.0) | 76.5 (22.5) | 84.0 (18.0) | 77.0 (23.0) |
| Mental *reflected self* | 88.0 (22.0) | 83.0 (22.0) | 91.0 (22.0) | 83.0 (24.0) |
| | $p < 0.001$ | $p < 0.001$ | $p < 0.001$ | $p < 0.001$ |
| | ES = −0.575 | ES = −0.457 | ES = −0.585 | ES = −0.508 |

**Note:**
All $p$-values significant with the Holm-Bonferroni correction; ES, effect size given by the matched rank biserial correlation.

Additionally, we tested whether significant differences between the *actual self* and other self-schema elements occurred within subgroups of the participants. Wilcoxon signed-rank tests were conducted for this purpose. Due to multiple comparisons, the Holm-Bonferroni correction was used: the number of planned comparisons was five within each of the four groups. The results are shown in Table 3.

In each group, significant differences occurred between the *actual* body image and its other aspects. The participants displayed their desire to have a slimmer figure and believed that their body type was ampler than both male and female standards concerning the most preferred body shape. When the participants compared their figures with the way other people perceived them, they indicated that their *actual* body image was almost in every case ampler than the *reflected* body image, excluding the group with excess BMI values. However, the comparison of the measured mental aspects of *self* showed that all groups of women were characterized by a more positive *reflected self* than *actual self*.

### H4: Self-reported BMI, weight-related stigmatization experiences, and the stereotype of a person with obesity

We hypothesized that women from different groups would demonstrate a similar way of perceiving people with obesity (H4). However, the Mann–Whitney $U$ test showed that women with excess BMI values had a more negative stereotype of people with obesity than women with normal BMI: this difference was significant at $p < 0.001$, and the rank-biserial correlation coefficient was 0.330; median (interquartile range) in the group with normal BMI was 2.40 (1.00) and in the group with excess BMI was 2.00 (0.85). Similarly, women with weight-related stigmatization experiences were characterized by a more negative image of people with obesity than respondents who did not declare such experiences: this difference was significant at $p < 0.001$ and the rank-biserial correlation coefficient was 0.227; median (interquartile range) in the group without weight-related stigmatization experiences was 2.40 (0.90), and in the group with weight-related stigmatization experiences was 2.20 (0.90). The median values related to the stereotype were in the lower range of the scale, which indicates negative attitudes toward people with clearly higher-than-normative weight in each study group.

## DISCUSSION

### Relations between self-reported BMI and weight-related stigma experiences

The analyses performed demonstrated that weight-related stigma experiences generally co-occurred with excess BMI. However, the number of respondents who experienced weight-related stigma was much larger than those who had a BMI above the norm. It suggests that the psychological significance of weight stigma diverges significantly from what above-normative weight means in an objective, medical sense (cf. *Cooper et al., 2007*). This discrepancy creates the risk that negative psychological phenomena accompanying the confrontation with the sociocultural context of weight stigma may accompany women with excess weight and a significant number of women who only think that they are above the normal range (*Romano, Haynes & Robinson, 2018*). Currently, on the one hand, body weight categories are used as indicators of other traits (*e.g.*, mental attributes). On the other hand, they determine the scope of activities which the individual can undertake without feeling ashamed or mismatched (*van Leeuven, Hunt & Park, 2015*). However, the frequency of weight-related stigmatization experiences among women whose weight is normal suggests that weight-related categories can serve as a social control tool (*Bartky, 1997*; *Rail, Holmes & Murray, 2010*). Its effectiveness may be all the greater because while other negative stereotypes (*e.g.*, racial or sexual) are currently clearly rejected by the majority of people, there is social consent for the negative treatment of individuals with excess weight (*Brewis, SturtzSreetharan & Wutich, 2018*; *Charlesworth & Banaji, 2019*).

### Body image, mental *self*-schemas, and self-esteem

Significant differences noted in the *actual* and *reflected* body images did not seem to be surprising. Women with excess BMI values perceived their figure as ampler and believed

that this was how other people perceived them (see Table 2). More interesting results were obtained as regards the *ideal* aspect of the body image. No significant differences were observed in this respect between women with and without weight-related stigmatization experiences. It supports the thesis that culturally created models of an ideal figure exist in today's world and are assimilated by most people regardless of their actual appearance (*Swami, 2015*), especially since the differences between the *actual* and *ideal* body image were also noted in other studies (*Crossley, Cornelissen & Tovée, 2012*; *Rozin & Fallon, 1988*). From a different perspective, women with excess BMI values indicated a slightly ampler figure contour as ideal compared to respondents with normal BMI. However, both groups of women agreed on what their bodies should look like to fit in with the preferences of both men and women. These results also supported the notion that universal models of attractiveness were internalized by the respondents, who compared their *actual* figure to them (*Engeln-Maddox, 2005*).

Interestingly, on the one hand, the figures indicated as preferred by men were slightly ampler than those which in the opinion of the respondents were preferred by other women. On the other hand, they were more similar to the figure types, which the respondents indicated as ideal for themselves. It suggests that the participants were more interested in conforming to the canons of beauty formulated within the framework of the other gender. An additional analysis performed in this context showed that according to the respondents, the standards of appearance imposed by other women were even more stringent than male standards (in Wilcoxon signed-rank test, the *p*-value was less than 0.001; matched rank biserial correlation = 0.807). Therefore, it seems that other women were seen as more severe "judges" than men in terms of physical appearance (cf. *Evans, 2003*). From an evolutionary perspective, this may mean that physical appearance is a signal of attractiveness to potential partners and a tool used in the intra-gender competition for a social position (*Fink et al., 2014*). However, *Crossley, Cornelissen & Tovée (2012)* indicated that male and female ideal images of the female body were compatible. Thus, according to men and women, the differences related to *ought* body images indicated by the participants may not be reflected in reality.

In most cases, higher numerical differences occurred between the figures indicated as those reflecting the *actual* body image and other aspects of the body image among women with excess BMI and women who experienced weight-related stigmatization. An exception was found in differences between the *actual* and the *reflected* body images, which were statistically significant but not pronounced in numerical terms based on median and interquartile range values. Thus, the discrepancies between the *actual self* and *reflected self* were most similar in the studied groups among all measured discrepancies.

The results related to the images of the mental *actual self* and the mental *reflected self* overlapped with those connected with body image aspects. However, after applying the Holm–Bonferroni correction, there were more significant differences in relation to weight-related stigmatization experiences. The respondents without weight-related stigmatization experiences had a more positive image of the mental *actual self* than those who declared such experiences. These women also demonstrated a higher conviction that their image was more positive in the eyes of other people. Further analyses showed

that all the respondents had a more positive image of the mental *reflected self* than the mental *actual self*, and the intergroup discrepancies between these two images were insignificant.

Women with normal BMI and without weight-related stigmatization experiences were also characterized by a higher level of self-esteem than women with high BMI values and those who declared such experiences. The obtained results are in line with the previous reports on the relationship between body weight and self-esteem in women (*e.g.*, *Miller & Downey, 1999*; *Kiviruusu et al., 2016*). However, the difference related to BMI values was insignificant after applying the Holm–Bonferroni correction. It suggests that how the respondents perceived their value might have been more related to their attitudes toward their bodies rather than what they looked like (*Lennon et al., 1999*; *Shahyad, Pakdaman & Shokri, 2015*).

Further analyses of the relationships between the *actual* body image and other body image elements showed that discrepancies between them were equally present in all study groups. The respondents consistently indicated that their figure significantly differed from their preferred body shape as well as the body type preferred by men and other women. The fact that even women with normal weight and without weight-related stigmatization experiences declared significant discrepancies between their actual figure and the *ideal* and *ought* canons related to it suggests how common dissatisfaction with one's appearance is (*Ghaznavi & Taylor, 2015*; *Homan et al., 2012*).

## The stereotype of the person with obesity

Given that the body is an element of culture that regulates the ways of using it and provides interpretations of its appearance (*Adelman & Ruggi, 2016*), we predicted that regardless of their body weight and weight-related stigma experiences, the respondents would have similar beliefs about people with excess weight. The study results revealed unfavorable stereotypes of people with obesity in the groups of respondents. However, respondents with high BMI and with weight-related stigma experiences expressed even less positive opinions. This result indicates the wide extent of sociocultural stigmatization of people whose weight is above the normal range (see *Vartanian, 2010*). Individuals with normal BMI and without weight-related stigmatization experiences convinced of the negative traits of people with obesity may feel satisfaction from being better than the above-mentioned group and may pursue their daily activities with a sense of anxiety that they might potentially contribute to being classified in the group with undesirable characteristics. In turn, individuals with excess BMI and weight-related stigma experiences who share the stereotypes about themselves may contribute to their consolidation and may not feel motivated to change them for two reasons. The first one is related to the hope of leaving the stereotypical group and becoming a member of the normative group (*Becker & Tausch, 2014*). The second one is connected with the gradation of the physical problem of excess body weight. Women with excess weight may perceive those who diverge from the norm even more as having a higher intensity of negative traits. Thus, paradoxically, the stereotypical perception of people with obesity may help them maintain higher self-esteem, as they feel that others are even worse than them (*Brown et al., 1992*).

## Study limitations

The study sought to consider factors that may be significant for the functioning of women with excess weight. Particular attention was focused on making sure that the respondents would constitute a homogeneous group of young adults to minimize the risk of physical differences between them related to developmental factors (*e.g.*, hormonal factors). Care was also taken to exclude individuals with excessively low body weight. Moreover, the complexity of normative aspects of weight was considered, including its actual aspect (normative criteria of BMI according to WHO) and the subjective experiences of weight-related stigmatization.

However, we found factors that could affect the obtained results. Although BMI is commonly used to determine whether an individual's weight is within the normal range, it also has certain disadvantages. One of them is the possibility of obtaining higher results by individuals who perform physical exercise or athletic individuals whose muscular figures can be classified in the same category as those of people with overweight (*Nevill et al., 2010*). Another disadvantage was related to applying a general category of excess weight in terms of respondents with BMI values ≥ 25.00, although a distinction between people with overweight (BMI between 25.00 and 29.99) and people with obesity (BMI > 30.00) would have been more specific here. We decided to adopt a general cut-off point to avoid excessive fragmentation of the groups. In the sample, women with excess BMI values accounted for 21.4% of the respondents, 14.2% of whom had BMI indicating overweight, and 7.2% had BMI indicating obesity. However, the analyses were conducted to test whether similar conclusions to those presented in the paper could be drawn if BMI could be treated as a trichotomous variable (as an indicator of normal weight, overweight, and obesity). The analyses are given in Table S1 (see Supplemental information). The obtained results are similar to those regarding hypotheses 2–4. Nevertheless, differentiation between subgroups exceeding the normative weight is recommended on a larger sample.

Studies also showed that BMI calculation based on declarations could lead to underestimating body weight and overestimating height (*Gorber et al., 2007*; *Maukonen, Männistö & Tolonen, 2018*). This effect may have been partly responsible for the study discrepancy between the size of groups with excess weight and weight-related stigmatization experiences. In turn, some research showed that study participants reported their weight fairly accurately (*Luo et al., 2019*; *Stunkard & Albaum, 1981*; for young adults, see: *Olfert et al., 2018*).

As the study was conducted on a sample using the snowball method, questions may arise regarding its representativeness and the possibility of generalizing the obtained results. The applied research methodology allows us to consider the obtained results preliminary. Further studies on more representative samples are warranted. From a different perspective, the percentages of women with normal weight and excess weight who participated in the study were very similar to those among young women in the general Polish population. According to the Statistics Poland data (2014; URL: https://stat. gov.pl/en/topics/health/health/percentage-of-persons-aged-15-years-and-more-by-body-

mass-index-bmi,11,1.html; accessed on 7 April 2021), among Polish women aged 20–29, 73.2% of people had normal BMI, and 17.8% had excess BMI, whereas among Polish women aged 30–39, 64.3% of people had normal weight and 31.5% had excess BMI. These percentages are comparable to those in the study (78.6% and 21.4%, respectively), especially if we consider that the general population also includes people with low BMI, and the difference in proportions would even decrease if they were not considered.

Doubts may also arise regarding the mental aspects of the *actual self* and the *reflected self* based on a single generalized value, constituting the result of the measurement using the Self Discrepancy Scale (SkRAP). This tool allows the comparison of responses with regard to each item. However, this would have led to the replacement of multi-item scales with good internal consistency with a set of single-item scales for which it would not have been possible to establish the reliability level. There are also limitations related to the determination of body image aspects based on the Contour Drawing Rating Scale. According to *Crossley, Cornelissen & Tovée (2012)*, methods based on a limited set of alternatives presented in 2D may have low validity concerning actual preferences related to body image. On the other hand, the advantage of the above scale is that it can be used in large sample surveys such as the one in this paper.

An important issue not addressed in the study is the interactions between self-reported BMI and weight-related stigmatization experiences. Such analyses were not performed due to the disproportions of participants that would have been distinguished in the 2 × 2 pattern. The group of women with excess BMI values who did not declare weight-related stigmatization experiences would have been particularly too small (only eight respondents).

There may be some concern related to the validity of the weight-related stigmatization method, which consists of two questions listed in the "Materials & Methods" section. The designed method was used only in the present study. However, it fits well with the assumptions of the concept of stigmatization, considering social and personal internalization aspects (*Goffman, 1986*; *Kahan & Puhl, 2017*). The question "Do you think you are fat now?" was considered a weight-related stigmatization experiences indicator since it refers to the introjected negative category related to body weight. In Polish, there are many euphemisms and synonyms for excess weight (cf. *Roszak, 2013*), and the meaning of the adjective "fat" is similar to English "thick" and considered rather insulting. Some studies also show that people described as "fat" were perceived more negatively than people described with other weight descriptors (*Brochu & Esses, 2011*; *Smith et al., 2007*) or without any weight descriptors (*Smith et al., 2007*). Using of the adjective "fat" as a self-descriptor seems to be consistent with *Goffman*'s *(1986)* concept of "spoiled identity", which was the theoretical context of the presented study. However, within the fat acceptance movement, attempts are currently being made to make the adjective "fat" a neutral descriptor (*Meadows & Daníelsdóttir, 2016*).

The question "Are you currently ashamed of how much you weigh?" was considered a weight-related stigmatization experiences indicator because it literally refers to the embarrassing aspects of weight-related stigma. The question "Are you currently ashamed of how much you weigh?" may seem problematic here because someone may be ashamed

not only of too high body weight but also of too low body weight. However, the probability of a significant distortion of the study results due to the above factor was minimal. Of all respondents, only 11 (2.9%) participants declared that their *ideal* body image was ampler than the *actual* body image. Among them, only one person confirmed embarrassment due to her figure and declared no feeling of "being fat" and no social signals indicating it. Similar doubts may arise because some excess weight people might perceive themselves as "fat" but not be ashamed of it, which could be an indicator of "fat and proud" self-schema (*Meadows & Daníelsdóttir, 2016*; *Casadó-Marín & Gracia-Arnaiz, 2020*). However, only seven of the 80 participants with excess weight responded to that pattern in our sample. Moreover, the additional analyses performed using the Mann–Whitney $U$ test, and the Holm–Bonferroni correction showed no statistically significant differences between these seven participants and the remaining 73 participants with excess BMI in terms of measured aspects related to body image, mental self, self-esteem, and obese people stereotype. However, future studies with a similar methodology should focus in a less ambiguous manner on weight-related stigmatization experiences.

## CONCLUSIONS

Regardless of BMI, any woman may experience weight-related stigmatization, which is related to failure to meet the standards of attractiveness (*Krzemionka-Brózda, 2010*; *Leksy, 2013*; *Swami, 2015*). The study results show the similarity between the psychological functioning of women with self-reported excess BMI and those who feel that their weight was above the normal range. The results also provide guidelines for practical actions aimed at reducing negative mental states associated with not conforming to the canons of beauty. First, the disproportion between the number of respondents with excess BMI and the number of women declaring weight-related stigmatization experiences indicates that weight stigma is only partly based on objective factors. Therefore, mitigation of its impact requires applying measures aimed at changes in psychological, social, and cultural areas (*Brewis, SturtzSreetharan & Wutich, 2018*).

Second, the discrepancies revealed in each group between the *actual* body image and its other aspects lead to the conclusion that dissatisfaction with the body shape is a common phenomenon, regardless of the actual compliance with the bodyweight criteria. The above should be indicated to individuals who struggle with excess body weight, who may experience disappointment when expectations clash with reality and weaken their motivation to pursue healthier habits (*Garner & Wooley, 1991*). From a different perspective, a longitudinal study by *Rancourt et al. (2017)* showed that weight misperception among young people with excess weight did not necessarily lead to further weight gain. However, the persistence of the positive effects of weight misperception may be difficult in the contemporary cultural context in which constant attention is paid to the importance of a slim figure.

Third, the results show the importance related to the opinions of others in the context of body normativity. The respondents unanimously declared that the ought body standards were more stringent from the point of view of other women than the corresponding standards attributed to men. The results suggest that women feel a stronger sense of a

mismatch between the models articulated within their gender. This aspect should be considered when developing strategies to eliminate the harmful effects of weight stigma. To replace narratives depicting men and women as opposing forces fighting for different standards, it might be appropriate to introduce communication showing that it is in the interest of all women to change the perception of the body shape, regardless of the size they wear (see: *Webb, Wood-Barcalow & Tylka, 2015*). From a different perspective, in future research, aspects of the perceived body image could be measured in a more controlled way if the respondents were asked to indicate their sexual orientation and to indicate what body image in their opinion is ought in specific spheres of everyday life (*e.g.*, sexual, work-related or social).

### Funding
The authors received no funding for this work.

### Competing Interests
The authors declare that they have no competing interests.

### Author Contributions
- Łukasz Jach conceived and designed the experiments, analyzed the data, prepared figures and/or tables, authored or reviewed drafts of the paper, and approved the final draft.
- Sonia Krystoń conceived and designed the experiments, performed the experiments, analyzed the data, authored or reviewed drafts of the paper, and approved the final draft.

### Human Ethics
The following information was supplied relating to ethical approvals (*i.e.*, approving body and any reference numbers):

The Ethics Committee of University of Silesia granted ethical approval to carry out the study (decision number: KEUS.18/04.2020).

### Data Availability
Data and codebook are available in the Supplemental Files.

### Supplemental Information
Supplemental information for this article can be found online at http://dx.doi.org/10.7717/peerj.12047#supplemental-information.

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
