# Peer review of "Self-reported body weight and weight-related stigmatization experiences among young adult women—two contexts, but similar attitudes related to body image, mental *self*-schemas, self-esteem, and stereotypes of people with obesity"

_PeerJ, doi:10.7717/peerj.12047_

## Round 0.1 · original submission · Major Revisions

First of all, my apologies for the slow reviewing/decision process. It was difficult to find suitable reviewers under the current circumstances, particularly around the end/start of year period. We were very fortunate to have two insightful and constructive reviewers, who have collectively made a set of excellent comments about your work. I feel that you could potentially address all of their comments, some of which I will briefly summarise below, along with my own additional comments further below, and so I will invite you to prepare a revised version of your manuscript along with a response to each comment (the reviewers’ and mine—including the reviewer comments I do not cover in my summary) where you address each one separately by explaining the changes you have made in response to it or by explaining why you have not made any change for that particular comment. I think that you will find their comments to be helpful, and hope the same for mine. I look forward to seeing your responses and reading your revised manuscript.

Both reviewers makes important points about definitions and terminology. Please ensure that all terms are defined in a way that would be understandable to a multidisciplinary audience. I feel that work such as yours can be very relevant to non-specialists and, in my own experience, is often picked up by the media, sometimes with resulting misunderstandings. I agree with Reviewer #1 about consistency, and I will also ask you to replace references such as “obese people” with people-first language such as “people with obesity”.

Reviewer #2 has been extremely helpful in noting some typos that have crept into your work and I will add that the wording is sometimes awkward in places. The manuscript needs careful editing by a fluent speaker of English. I’ll note that PeerJ, and several other organizations, provide editing services should you wish to pursue that option. Both reviewers have made other suggestions aiming at making your manuscript more readable, and I find myself in agreement with Reviewer #2 about the length of the introduction. If it is possible to shorten this, at least a little, without losing the important (and interesting) content, I think readers will appreciate this. They also note that it is conventional to state the aim(s) of the study in a paragraph at the end of the introduction, i.e. around Line 141 here, which will help readers accustomed to this convention; in the present manuscript, this is currently covered in Lines 144–149 so it’s not the order that they are querying, I think, but the structure and reader expectations concerning this. However, note that the study aim as written does not seem to explicitly refer to young women, or even women, which should be made clear there. While there are different conventions in different fields, I also agree with Reviewer #2’s point about needing a data analysis section, bringing together: the summary statistics used, the inferential statistical models used, the model/test diagnostics and any related decision-making process for these models/tests, the treatment of missing data, the software used, and the level of statistical significance and any treatment of multiple comparisons. This will help to make the analysis approach clear to the reader (if you have code for the analyses, that would make an excellent addition to the online materials) and should provide sufficient detail so that a reader who has a copy of the data could expect to replicate your results without resorting to trial and error (which is where code can be particularly useful). To these points, I’ll add that it might be more natural for your sample description (Lines 198–207) to go in the Results section with a traditional Table One showing all of the relevant descriptives that you are able to present (I’ll also suggest that a single decimal place is sufficient for age, c.f. Line 202, and that both counts and percentages be given in some way [text and/or table], c.f. Line 203, for example; given that each women contributed over 0.25%, the second decimal place for percentages, e.g. Lines 515 and 516, also seem a little excessive to me). It’s best to avoid interpretation (e.g. “Interestingly” on Line 278) in the Results section and instead leave such for the Discussion.

It is crucial that a revised version addresses the writing of the manuscript so that readers can follow and appreciate your very interesting work.

Reviewer #1’s question about the BMI cut-off is important, particularly given that mean BMIs in many countries, including Poland, are above 25 for women (and also men). I will suggest that you explore a higher cut-point in addition to 25 to see if your overall conclusions are robust to this change, but I would be very happy to hear reasons for why you think this shouldn’t be done. Note that finding a dose-response association where there was progressive divergence (or convergence) going from “people with normal weight” to “people with overweight” (n=53) to “people with obesity” (n=27, still sufficient for comparisons given that 95% CIs will quantify the precision of those comparisons— I appreciate the issues with small sample sizes precludes you addressing questions about two-way analyses involving both self-reported BMI and self-reported stigmatization) would present much stronger evidence for an association than a two group comparison. An alternative analysis strategy might have involved keeping BMI as a continuous measure and looking at its association with the outcomes of interest here using linear (or quantile) regression.

Reviewer #1 helpfully provides a reference about weight misperception that might add to your discussion. You are, of course, free to use this or (an)other reference(s) if you do decide to incorporate this into your manuscript. Reviewer #2 asks about references to previous uses of this procedure. The validity of your stigmatization construct is clearly very important, as Reviewer #2 suggests, and you need to establish this validity as best you can. The third of the stigmatizing questions (“Are you currently ashamed of how much you weigh”) seems to me to be slightly ambiguous in that being underweight could also lead to an affirmative response. I’d be surprised if this happened often, if at all, but it does seem a point that some readers might wonder about.

Both reviewers ask some important questions about what can and cannot be concluded from your study and Reviewer #2 asks you to more clearly identify the gap in the literature you are addressing (but note that novelty is not a requirement for PeerJ and meaningful replications are also interesting).

Reading the English version of the participant consent sheet, and thank you for being so kind to provide this, I noticed that it said: “and is addressed to every WOMAN [sic] over 18 years of age.” However, in the manuscript, you say that you dropped data from 45 participants as they were aged over 35 years of age (Line 200). This is a reasonable proportion (~10% of the initial sample on Line 198) of your dataset and a large number of people who have spent time on your study (which would have ethical implications as well as statistical internal validity and external validity ones). Can you explain why there were so many people in this age group recruited into your study and why they were dropped from analysis here? Are they being analyzed in another manuscript?

Perhaps related to this, I’d like to know more about your recruitment. Which social networking sites (Line 189), how were they invited/what did they know about the study (Line 191 is a little vague about “the purpose of the study” for me), and how were they asked to pass on the invitation to other people (how were these other people described, Line 189)? Can you be clearer about the 3 participants with incomplete data (Lines 200–201)? What was the minimum set of data you required for them to be included here?

Some of the descriptive statistics suggested some skew in the data, and as you would expect based on Line 202, age is positively skewed and might be better described using a median and IQR instead of an arithmetic mean and SD (or a geometric mean and SD could be used if the data was approximately log-normally distributed). You should check all other reported means and SDs for the same point. This process of deciding how to describe your measurements can form the first sentence or two of your statistical methods section as described above.

The largest threat to your study’s internal validity seems to me to be the use of self-report height and weight for calculating BMIs. This is not an unusual approach and there are several articles available that investigate the validity of these self-reported values, some of which you cover in the discussion on Lines 520–526, but there are more recent references supporting claims that women (at least of some age groups) report their weights and heights reasonably accurately. I think you need to convince your reader in the methods though that any bias is unlikely to be too large here, and so the BMI categories are reasonably valid overall, before they read the results. Related to this, only 21% of your analytic sample being people with overweight or obesity seems very low, and still seemed low even after I looked into overall Polish data for overweight and obesity. How does this compare to Polish data that best matches your sample (do you appear to have either an under- or an over-representation of people with overweight/obesity compared to a simple random sample of your target population?) If your data differs from that expected, does this reflect your sampling strategy, an issue with self-report here, or simple chance? Can you explain this for your reader’s benefit and identify any biases and their likely effect on results that you consider plausible? When you refer to “actual body weight”, e.g. Line 267 but also elsewhere, I think this should be qualified as “self-reported body weight” so the reader keeps this potentially important point in mind when considering the results.

I’d like to see a “cleaner” presentation of the results in the tables. For example, the current Table 1 (which I think would become the second table after a conventional first table of participant descriptives) could be simplified to describing a statistical test in the text. Table 2 could similarly be described in a sentence in the text. Unless you are interpreting the degrees of freedom or test statistics in these tables, and also Cramer’s V which didn’t seem mentioned outside of Tables 1 and 2, you can feel comfortable leaving them out. The data will be available to readers so they can check your calculations if they wish. The test statistics and degrees of freedom in Table 3 are similarly unnecessary at the moment, as are those on Lines 424–425. Table 3 should, however, add 95% CIs, either for the standardized effect sizes, which you do make use of in the text, or for the raw effect sizes, and show actual p-values rather than asterisks. It’s also useful to indicate in a table note where the statistics come from for readers who skip over the statistical methods or revisit the manuscript without refamiliarizing themselves with the statistical methods. The same points as mentioned above (check means/SDs are appropriate, report differences with actual p-values and 95% CIs for standardized and/or unstandardized effect sizes) should also be addressed for Table 4. Table 5 again looks as if it could be described in a sentence in the text (could I also ask you to check the figures here as I was unable to replicate these using the “OPS_GS” variable in your data, which may reflect a misunderstanding on my part). You should feel very welcome to advocate for different approaches to presenting your data than I’ve suggested above.

Note that the addition of 95% CIs would allow you a more nuanced examination of some of your questions. For example, when you report statistically significant evidence for differences in obesity stereotypes for women with overweight/obesity (Lines 369–371), the p-value is much less important than the estimated mean difference (which could be standardized) and the 95% CI limits in indicating whether or not this difference is likely to be of practical importance. Similarly, non-statistically significant results might include or not include practically important values in their 95% CI which will give some guidance as to whether future research is or is not recommended to better estimate these effects.

When you compare the number reporting stigmatization against the number reporting overweight/obesity, this is a paired analysis (there is error/imprecision on both sides and neither measure is known with the necessary certainty to allow for a one-sample test against a fixed/known value) rather than the one-sample test of proportions you report on Lines 267–271. McNemar’s test would be appropriate in this case for the paired binary data (stigmatization and overweight/obesity).

Based on a quick look at the data, I’m not convinced that the assumptions around t-tests are sufficiently well satisfied in all cases and, if you agree, you might wish to look at non-parametric tests (such as Mann-Whitney U), which would have the disadvantage of losing the 95% CIs mentioned above. You should also check the assumptions behind the paired t-tests (Line 346). Both sets of statistical model/test checks should be described in the statistical methods to reassure the reader that these necessary tasks were performed.

While I appreciate your use of Bonferroni (Lines 298–300 and again Line 347-349) to accommodate the multiple comparisons given your confirmatory, as opposed to exploratory, approach, this would be overly conservative here due to the non-independence of the tests. You can retain this for simplicity if you wish, but alternatives such as Holm–Bonferroni would preserve the overall family-wise error rate while reducing the risk of type 2 errors. As a starting point, https://en.wikipedia.org/wiki/Holm%E2%80%93Bonferroni_method provides an overview of the method and its motivation.

As a last statistical suggestion, your aim on Lines 144–147: “to compare the strength with which two aspects related to body weight embedded in objective factors (actual BMI) and experienced more subjectively (weight- related stigmatization experiences) influence body image, mental self-image, self-esteem and stereotypes concerning people [with obesity]” would be most naturally addressed using a linear mixed models (often referred to as hierarchical linear models in psychology) depending on assumptions being met. This would allow you to look at the within-subject differences in the relationship between body weight (two rows per participant, self-reported BMI and self-reported stigmatization) and the outcomes of interest and so ask whether the associations were “stronger” in some sense for one source of “body weight”. Repeated measures ANOVA would be another option, but this wouldn’t allow some extensions alluded to in the following paragraph to further reduce the total number of statistical models used.

Related to this, the notion of “stronger” is problematic in statistics, e.g. Lines 326 and 328 here, as it is ambiguous as what it would mean (neither of smaller p-values or larger, raw or standardized, effect sizes entirely satisfies this description) and so I would suggest rewording any instances where this word (“stronger”, or “strength” in a related context) is used to describe statistical analyses and/or results. The text on Lines 325–329 could, however, be better addressed with mixed model analyses as described above as you would then be directly comparing effect sizes between the comparison using each of the two “body weight” measures. (Such as an approach would also allow looking at the paired analyses in Table 4 at the same time, including the associations involving male and female ought body images.) In the absence of this, and I’d entirely understand if you didn’t wish to go down this analytical path, you could still note, if it was the case that there were more statistically significant findings using one such measure, or that effect sizes had a pattern of generally being larger for one measure, or that the differences were similar (as you do on Line 303 and elsewhere) but you should be clear whether this is in terms of effect sizes, statistical significance, both, or something else. Numerical comparisons are fine, as long as the reader will always appreciate that this is what is intended by the text. The addition of “numeric”, “numerical”, or “numerically” can be useful in making this distinction clear in my experience.

In terms of the presentation of results, I did wonder if Table 4 might not work better before Table 3, the discrepancies in Table 3 compared between groups are after all investigated in Table 4 within groups, but I’ll leave this up to you.

Stylistically, I wondered if “ought” ought to be italicized or somehow otherwise indicated as special in the manuscript. You do this on Lines 101 and 105, and could continue this throughout the manuscript as it has a more technical meaning than just the ordinary English meaning. You could do the same for “ideal” and “actual”, again as you do on Lines 100–110, to remind the reader that these words also have technical meanings here. This pops up again at least once later, e.g. Line 345, so I’m just suggesting a consistent approach throughout the manuscript.

·

Basic reporting

a. In opening paragraph of intro (Lines 52-63), you should provide a basic definition of what you mean by weight stigmatizing experiences. People in higher body weights can experience in a variety of settings (medical, relationships, weight related teasing, etc)
b. Some inconsistencies in terminology (noticed especially in paragraph lines 84-97). You use “higher body weight” and “obese or overweight.” There is not consensus on which term to use. The medical community leans towards “obese,” and yet many higher body weight people find that language stigmatizing. Research what term you would like to use and stay constant.
c. Overall, authors have provided a well-researched background for the topic. However, the overall structure and order of the paragraphs was slightly difficult to follow at times.
d. Raw data shared

Experimental design

a. Curious as to why you used BMI above 25 as your cutoff. You may see different results if you raised your cutoff to 30 or higher. I think this cutoff needs to be supported further. You comment on this a bit in discussion, but I am not convinced this was the correct cutoff

Validity of the findings

a. Line 566 – You make a conclusion that getting rid of weight stigma is not tantamount to positive outcomes. I am not sure you can draw this conclusion from this type of study design.
b. You may find it beneficial to your argument to reference the weight misperception literature. That is, the literature that shows women who are higher weight but perceive their weight has normal have better health outcomes. (Sonneville, KR)

Additional comments

a. Overall, adequate review of literature, interesting study questions, clear methods. I would like to see some restructuring of background for clarity, and other comments – but interesting addition to the weight stigma literature

Reviewer 2 ·

Basic reporting

(a) Abstract:
* The comma is missing between "mental self-schemas" and "self-esteem":
“The paper compares body image, mental self-schemas self-esteem and the obese person stereotype among groups (…)”.
* I do not understand this sentence “Importantly, the studied women indicated that the ought body image from the point of view of other women should be slimmer compared to the male point of view.”
* As for the conclusion, do not these parallels apply to women with excessive body weight and stigmatized women?

(b) Introduction:
* What do "higher body weight" and “higher-than-normative body weight” mean? It should be clearly defined what these terms mean for the authors.
* The Introduction section and the presentation of the aims and hypotheses takes up 6 pages. In my opinion, this part is way too long. Authors should try to present this information in a more synthetic way - without duplicating the same information, focusing on the results of research related to their research and selecting the latest research results.

Experimental design

(a) The purpose described in the Abstract and the Method section is not consistent. Moreover, “Aims of study and research hypotheses” should be presented at the end of the Introduction, not in the Method section.

(b) Unfortunately, it is not entirely clear to me into how many groups were the women ultimately divided? Were women divided taking into account both their body weight (normal body weight or excessive body weight) and whether they felt stigmatized (yes or no) (i.e. ANOVA 2 x 2)?

(c) The manuscript should provide specific BMI ranges from which the division into "normal" and "higher" has been made with reference to the source (WHO?). Moreover, I would suggest changing the concept of "higher" to "excessive".

(d) Have other researchers used such a procedure before? I am not entirely sure whether the answer to the question "Do you think you are currently fat" clearly indicates that someone feels stigmatized.
“The following questions were asked: “Do you currently happen to hear anyone say you are fat?”; “Do you think you are currently fat?”; and “Are you currently ashamed of how much you weigh?” If an affirmative answer was given to any of these questions, the studied women were assigned to the group characterised by weight- related stigmatization experiences."

(e) It is not clear to me how this study fills the knowledge gap

Validity of the findings

(a) Abstract:
“Higher-weight women and women with weight related stigmatization experiences were characterized by greater divergences between the actual body image and the ideal, reflected and ought body image, lower self-esteem and more negative beliefs about their mental actual and reflected self”…. compared to?

(b) Method:
* There is no "Data analysis" section in which the authors will present what statistical analyses they used.
* If I understand correctly, the authors assumed that both body weight and the experience of stigmatization are important factors for the variables they analyse. So, why was two-way ANOVA not used and all 4 groups not compared (this remark obviously applies to quantitative variables)?

Additional comments

Thank you for the opportunity to read your work. This is an interesting topic that can be considered by readers. Nevertheless, there are some concerns with the present write-up that would need to be addressed for the paper to be able to achieve its potential.
Though the paper is properly organized, it is not and easy to follow. Some issues are not clearly defined (e.g. group division), which makes it difficult to understand the various parts of the manuscript.

---

## Round 0.2 · Major Revisions

Thank you for your revisions and responses. I appreciate the constructive approach you have taken and the extensive work you have done in response to the comments from our two excellent reviewers and from myself. I feel that your manuscript has been definitely improved through all of your hard work. One of the original reviewers has commented on this new version of your manuscript. While their comments are brief, they raise some substantial issues that need to be addressed.

While I agree with the reviewer that the introduction remains long, your research is also broad in scope and likely to be of interest to a multidisciplinary audience (as I suggested previously). While I’ll ask you to look for opportunities to tighten this section, I appreciate that you might not be able to reduce it by enough to achieve a “normal” length introduction here.

You’ll see also that they remain unconvinced about the items about stigma. I appreciate your desire to further examine the properties of these items in the future and your arguments added to the discussion about the “shame” question are helpful. The “hearing another person describe them as fat” item might also warrant a mention here in the discussion as this could, perhaps, represent more general bullying in some instances. The “current perception as fat” item could also be addressed here as this doesn’t necessarily represent stigmatisation to someone with body positivity (would someone who says, and genuinely means, “fat and proud” be stigmatised?) If you can cover all three items to a similar extent, I think that this might go some way towards assuaging some of the reviewer's concerns (but will ask them to share their thoughts regarding any revisions you make here). You could try to argue for their face and content validity. Or perhaps you could identify similar items (or scales) in the literature that have been validated. Another alternative might be to explore their meaning through (additional) study, such as focus groups. You might well have other ideas, but I do share the reviewer’s unease about exactly what these items are eliciting from the respondents and I think readers will appreciate you giving a similar amount of attention to all three items given how crucial they are to your research.

You’ll notice that they have also asked about the knowledge gap being addressed and there did not appear to be any text under your heading (in your response letter) “(e) It is not clear to me how this study fills the knowledge gap”.

I appreciate the addition of the statistical methods overview (Lines 240–245). At the risk of pedantry, I’ll note that the assumptions behind parametric statistical approaches that would address similar/related questions (e.g. two sample t-tests, one-way ANOVAs, and paired t-tests) aren’t about the data as such (which is what is described in Table 1, see Line 241), but rather concern the error term which can be explored empirically by looking at model residuals. For the first two of these analyses (two sample t-tests and one-way ANOVAs), the residuals will have the shape of the data WITHIN groups, but not the same as the data OVERALL. For the last of these (the paired t-test), the assumption for a paired t-test include that the DIFFERENCES are normally distributed, not the two sets of data themselves. This means that it is not possible to check the assumptions for many statistical tests/models (including paired t-tests, but also ANCOVAs, RM-ANOVAs, etc.) without (effectively) fitting the model so that the model residuals are able to be examined. I would normally recommend a visual process for this as formal statistical tests become overly sensitive to evidence for departures from assumptions in larger samples (where the departures matter least when the central limit theorem is in play) and are not sensitive to large departures in small samples that ought to leave one unsatisfied that we have evidence FOR the assumptions being met, a problem that also follows from testing for evidence AGAINST the assumptions being met—absence of evidence is not [strong] evidence of absence. To ensure that the reader could understand the choice of statistical methods, and to in part address the reviewer’s 5th comment, you need to explain how the statistical methods were selected, including these diagnostics. It is often easiest, in my view, to fit the parametric model, which you would generally prefer given its (often) greater statistical power and its greater interpretability; then to assess whether those models are adequate in terms of these assumptions; and then finally to switch to non-parametric alternatives when the assumptions are not well met (while bearing in mind that the questions are related but not the same for the alternatives here). There are some limitations to this approach that are discussed in the statistical literature, but I think it works well enough in practice for it to be used in cases such as here. Note that I’m not suggesting that you present any new information in the results, this would all form part of the statistical methods section. Two of the tests used at the present moment, McNemar’s and the signed Wilcoxon, are (assuming the residuals don’t support paired t-tests for the latter) needed due to the paired nature of the data (relating to the independence assumption) and this could be made clearer here also. One final point on the statistical methods text is that the adjustment for multiple comparisons needs to specify at what level this was performed (reading only the statistical methods, the reader could wonder if this was within each variable for post-hoc comparisons, for a set of models/tests, or within the manuscript as a whole) in the methods and not only later (e.g. Lines 262–263).

For Table 1, I think it would be useful to check that you believe the reader needs to know each of these statistics. As noted before, if you’re not referring to values in tables, it is often possible to leave these out entirely. I appreciate that in some fields, such as psychology and education, there is a tradition/history of reporting test statistics and degrees of freedom that is not found in all other disciplines, but I would recommend cleaner/simpler presentations unless you have a preference for reporting these yourselves. For example, could you simply state in the methods that due to evidence against normality, you’ve presented medians and 25th and 75th percentiles rather than means and SDs as descriptive statistics (potentially saving up to six columns from the top part of Table 1)? I’ll also suggest you think about the decimal places, three for age, as an example, seems difficult to justify to me—see my previous suggestion that a single decimal place would suffice here. There is what seems to be a clear error in median weight (6.000) and note that the units (kgs I am assuming) should always be made clear.

For Tables 2 and 3, while you can report means (and even SDs as were included in Table 3) alongside non-parametric tests, it is much more common to show medians (with either IQRs or the 25th and 75th percentiles). Where the assumptions of t-tests, etc. are not satisfied, this generally (but not always) calls means and SDs into question as the best measures of location and dispersion. Similar to above, are the Z-statistics useful for readers? If you feel they are, then they should remain, but in that case, I’d expect some interpretation of them to be included in the text. The same applies to the mean ranks in Table 2. Again, I’ll ask you to think about the decimal places as three seems excessive to me (I’m happy to hear reasons if you feel this is necessary, but if it’s possible to present [effectively] the same information with fewer digits, your tables will be easier to read).

A few other points that I noted on reading this version of the manuscript:

Line 248: You should explain the “B/C” in the statistical methods above.

Line 250: As noted previously, the decimal places for percentages seem excessive. Here, each participant contributes 1/80=1.25% in themselves making the use of three decimal places here seem unjustified. I’m happy to hear an argument for showing more decimal places, but I suggest a general rule of using integer percentages when the denominator is no more than 100, and one decimal place for denominators above 100 but < 1000.

Line 252: Note Chi-squared statistic here. Does this provide information for the reader?

Lines 249–254: I think the logic here needs to be made a little clearer to the reader, namely that the evidence of an association comes from McNemar’s test and the evidence for non-specificity comes from the descriptives (note the text on Line 253 could perhaps be misread as saying that both follow from the formal test).

Line 324: Note Z-statistic here also. See also Lines 325–326, 328, etc.

Reviewer 2 ·

Basic reporting

1. The introduction is still too extensive.

Experimental design

2. Of course, I understand that it may be the first time that we use a given measurement method in a given study. However, I still doubt whether this is a question can measure the variable that the authors wanted to evaluate.

3. It is still not clear to me how this study fills the knowledge gap.

Validity of the findings

4. Incorrect markings appear in the tables (e.g.!). Most often, we use * to describe p.

5. If variables are measured on quantitative scales, why are mean ranks calculated and reported for them?

Additional comments

Thank you for the opportunity to review a revised manuscript. I appreciate the authors' effort in responding to all comments. However, I still have some concerns.

---

## Round 0.3 · Minor Revisions

As you will see below, two reviewers have provided their comments on the latest version of your manuscript. Reviewer #2 has no further comments and Reviewer #1 had only a few comments that shouldn’t be too difficult for you to address. I think we are very close to the final version of your manuscript.

I think I can see what they mean about the stigma <--> weight gain cycle (Lines 64–65) and the weight loss --> less stigma link (Lines 66–68). Some careful rewording should be used here if you agree. Related to these points, they have asked about the association between “being” fat and experiencing stigma. I appreciate the considerable difficulties of developing an item or set of items that taps into this particular construct (weight-related stigma) and not others, and all of the work you’ve already done here, but I will ask you to consider their suggestion of further drawing the reader’s attention to this point and your data on it. They also ask for some more clarity around the inclusion/exclusion criteria. Perhaps around Line 152, you could identify these (being women and aged between 18 and 35 [inclusive] are identified on Lines 22–23, 113–114, and 165–167 but were there any others around physical health, language, etc.? If not, stating these around Line 152 along with the lack of additional criteria might anticipate readers wondering if, for example, women in prison or transwomen were eligible.) Their other comment was about a minor typo. If you can address these last comments from Reviewer #1 and those from me below, I will be delighted to accept your interesting manuscript and look forward to seeing it added to the literature discussing this important topic.

Lines 89–90: “they self-value”; do you mean “their self-value”?

Line 153: Sorry for not asking about this before, but on re-reading the manuscript, I wondered what other social networking sites were used. Would it be possible to give a couple of additional examples of the sites you used here? If the sites are sufficiently few in number, would it be possible to exhaustively list them here? I think the brief description on Line 25 (Facebook only) is fine, but I am curious as to what others were also used.

Line 189: Now that there are two items only, you could say “either of these questions” (rather than “any”).

Lines 197–199: I’m absolutely NOT asking for any new analyses here, but I wonder if sexual attraction would be a useful variable to consider in future research to see how/if this modifies any differences between the two ought body images, and any discrepancies between these and actual or ideal body images. This could connect into the “competition” referred to on Lines 379–381. Would your observation on Line 370–372 be expected to be exactly the same for those with opposite-gender versus same-gender attraction (and versus both or neither) and would that help to disentangle the social versus sexual components of “ought”? Sorry, this is just a random thought that might, or might not, be worth keeping in mind in the future and one that it seems very likely you’ve already considered.

Line 227: You could remove “B/C” and just say “McNemar chi-square test” here (as you do on Line 238) as the test always compares the B and C cells.

Line 239: “for matcheD pairs”

Lines 239–240: Perhaps “A TWO-SIDED p-value below…” here so the reader knows that you were not performing one-sided tests? (I am assuming you have used two-sided tests throughout.)

Line 248: Perhaps “THE McNemar chi-squared test…” (adding “the” and removing “B/C”).

Line 270: “Significant differences were also NOTED between women” (or “observed”, “seen”, or similar).

Line 303: Double word: “experiences experiences”.

Line 326: Did you mean the second value in “2.00 (.825)” to be to 3 decimal places? Elsewhere on this page, both medians and IQRs are two 2 decimal places.

Line 331: Did you mean “mean” here, or perhaps “median” instead?

Lines 456–457: The first percentage is given to one decimal place (21.4), which would be my suggestion here, but the next two are given to two (14.17 and 7.22), which seems overly precise to me.

Line 480: Again, two decimal places for some percentages. C.f. one decimal place for Lines 251, 253, 456, (just above) 478–479, and 512, and also Table 1.

Line 486: I suggest deleting the “a” in “with a good internal”.

Table 1: While it’s “obvious”, I’d add units of “years” to Age, e.g. “Age (years)”.

·

Basic reporting

Overall, background is sufficient and clear. I do have one comment about lines 64-68. I believe the authors have excellent intentions with this paper and I want to commend them on this paper, but I want to point out that there is underlying stigma about weight in this section. 1) While weight stigma is associated with additional weight gain, we want to be really careful of how we frame this argument. It perpetuates the idea that weight gain is inherently bad. I would agree that additional weight gain has the potential to be problematic, but its still important to carefully craft that argument. I think the next sentence should be reworded as well as saying that someone feels inferior despite the weight loss makes it sound like they should no longer feel inferior based solely off of the weight loss.

Experimental design

a. Be clear on your inclusion/exclusion criteria.
b. I have doubts about the question “Are you fat now,” as an indicator of weight stigma. You are making an assumption that an individual experiences dissatisfaction about their weight status. The other question seems adequate. I do think it may be worthwhile to analytically look at whether or not to include this question as a measure of weight stigma. (I did see that there is a section in the discussion section about the negative stereotypes held by your respondents regarding excess weight. If you choose to keep both questions, I think you could and should highlight this data as it does appear to support your assumption.)

Validity of the findings

a. Line 377 minor typo with p- value missing decimal point
b. I thought the edits that were added to the discussion were excellent and greatly improved your discussion.

Additional comments

Thank you for contributing this paper to the weight stigma literature.

Reviewer 2 ·

Basic reporting

No comment

Experimental design

No comment

Validity of the findings

No comment

Additional comments

Thank you for the opportunity to review a revised manuscript. I appreciate the authors' effort in responding to all comments, and I believe that most of the issues have been appropriately addressed. Well done!

---

## Round 0.4 · accepted · Accept

Thank you very much for your excellent rebuttal and your revised manuscript. I am absolutely delighted to accept your work and look forward to seeing it add to, and generate new debate in, this crucial area. Well done!

I’ll make a few comments below that you could take into consideration when finalising your manuscript before publication but I’ll leave these almost entirely to your discretion (the missing year on Line 69, if I’m reading that set of references correctly, should be added though).

Line 69: There seems to me to be a year missing after the citation “Myers, Rothblum” (2010 from Line 701).

Lines 84–86: This only struck me when reading this the latest time, but “A difference between the actual self and the ought self generates a feeling of failure to meet the standards associated with shame…” would seem to me to only apply in one direction for it to be a feeling of “failure”, i.e., where the “actual” self falls short of the “ought self”. While I suspect that this is the overwhelming direction for discrepancies, it seems possible for someone to feel that their “actual” self surpasses their “ought” self. Would it make sense to qualify this, e.g. as “A NEGATIVE difference between the actual self and the ought self generates a feeling of failure to meet the standards associated with shame…”? Even if you agree with my comment here, I think Lines 86–87 are fine as they are about a mismatch without a direction implied.

Lines 88–89: With the same idea in mind, though, perhaps: “However, a NEGATIVE mismatch between the actual self and the ideal self is connected with a depressed mood and a feeling of self-disappointment…”—as long as you believe it is possible for the “actual” self to surpass the “ideal” self.

Lines 113–114: I think there might be a spurious space after the dash in “(aged 18- 35)”.

Line 178: I think all of the text and most of the table results report the interquartile range (IQR, e.g. Lines 330, 334, etc. along with Tables 2 and 3) rather than “the lower and upper quartiles”, which do appear in Table 1. This text could be changed to “the lower and upper quartiles, OR THEIR DIFFERENCE, NAMELY THE INTERQUARTILE RANGE”.

Line 226: Is this pair of terms opposites: “setting limits for themselves/self-restricting”? These seem closer to meaning the same to me on re-reading and I wondered if a word might be missing.

Line 357: Perhaps I’m being pessimistic while re-reading your excellent work here, but “are currently clearly rejected,” seemed perhaps to need some qualification such as “are currently clearly rejected BY THE MAJORITY OF PEOPLE,”

Table 1: I appreciate that this is obvious, but I’d still add units for age, e.g. “Age (y)” or “Age (years)” depending on your preference. You already do this for height and weight.

Table 1: I wonder if the qualitative variables might not be better as a separate table. Either way, I think changing the order of the headings to “Yes”, “% Yes”, “No”, and finally “% No” might be easier to follow.

Table 1: Perhaps the comment “The group of participants…” could be made a note under the table?